# Bending Experiment and Mechanical Properties Analysis of Composite Sandwich Laminated Box Beams

**DOI:** 10.3390/ma12182959

**Published:** 2019-09-12

**Authors:** Xiujie Zhu, Chao Xiong, Junhui Yin, Dejun Yin, Huiyong Deng

**Affiliations:** Department of Artillery Engineering, Shijiazhuang Campus, Army Engineering University, Shijiazhuang 050003, China; sduzxj@163.com (X.Z.); yuanzhidao@163.com (J.Y.); dj3500560897@163.com (D.Y.); deng-124@163.com (H.D.)

**Keywords:** composite sandwich laminated box beam, three-point bending, ultimate load, stiffness performance, deflection analysis

## Abstract

The failure modes, ultimate load, stiffness performance, and their influencing factors of a composite sandwich laminated box beam under three-point bending load are studied by an experiment, finite element model, and analytical method. The three-point bending experiment was carried out on three different core composite sandwich laminated box beams, and the failure modes and bearing capacity were studied. With the use of composite progressive damage analysis and the core elastoplastic constitutive model, the finite element model of the composite sandwich laminated box beam was established, and the three-point bending failure process and failure modes were analyzed. The analytical model was established based on the Timoshenko beam theory. The overall bending stiffness and shear stiffness of the composite sandwich laminated box beam were calculated by the internal force–displacement relationship. The results show that the composite sandwich laminated box beam mainly suffers from local crushing failure, and the errors between the finite element simulation and the experiment result were within 7%. The analytical model of the composite sandwich laminated box beam can approximately predict the overall stiffness parameters, while the maximum error between theoretic results and experimental values was 5.2%. For composite aluminum honeycomb sandwich laminated box beams with a ratio of span to height less than 10, the additional deflection caused by shear deformation has an error of more than 25%. With the ratio of circumferential layers to longitudinal layers increasing, the three-point bending ultimate load of the composite sandwich laminated box beam increases, but the ratio of the overall stiffness to mass reduces. The use of low-density aluminum foam and smaller-wall-thickness cell aluminum honeycombs allows for the more obvious benefits of light weight.

## 1. Introduction

Sandwich structures are formed by two layers of panels and an intermediate core material through a glue layer. The panels are generally selected from a resin-based fiber-reinforced composite material having a high specific modulus and specific strength, and they can withstand large bending normal stress; the core materials generally adopt lightweight materials such as a honeycomb structure, grid structure, and low-density foam to fix, transfer load, and resist shear deformation [1,2]. A laminated box beam with composite sandwich structure wall panels can not only exert the advantages of the box beam such as good bending resistance, stable bearing on the side, and availability of internal space, but also achieve the purpose of reducing weight and improving shear resistance. As a kind of lightweight structure with superior comprehensive performance, composite sandwich laminated box beam structures are widely used in aircraft wings, helicopter rotors, wind turbine blades, bridges, and many other large structures. Studying the stiffness performance, ultimate load, and bending failure mechanism of the composite sandwich laminated box beam structure can provide guidance for structural design and engineering applications of composite sandwich laminated box beams.

A large number of studies were carried out by scholars on the mechanical properties of composite sandwich laminated box beams. Mai et al. [3] established an analytical model for the overall stiffness of a three-point bending laminated box beam considering shear deformation, analyzing the structural response of a composite box beam with a wall filled by aluminum square honeycomb, corrugated board, and foam; the relationship between mass and overall stiffness was studied. Wenming et al. [4] calculated the bending stiffness of a composite aluminum honeycomb sandwich laminated box beam using Euler–Bernoulli classical beam theory, and analyzed the shear lag effects of flanges through finite element analysis. Qiang et al. [5] studied the bending response and failure process of a carbon fiber reinforced plastic square tube filled with aluminum honeycomb by three-point bending experiment and finite element analysis.

In summary, the existing analytical models of composite sandwich laminated box beams are based on classical beam theory, without considering non-classical effects such as lateral shear deformation, warping, and three-dimensional strain. There is no overall force analysis in this method, which reduces the accuracy of calculation. It is also rare to study the three-point bending ultimate load and failure modes of composite sandwich laminated box beams using finite element simulations. In this paper, three-point bending experiments were carried out on composite sandwich laminated box beams of three different core materials. The failure process, overall bending stiffness, and ultimate load of the three different core composite sandwich laminated box beams were compared. The progressive damage analysis model of the composite material and the elastoplastic constitutive model of core material were introduced, and the three-point bending finite element models of the composite sandwich laminated box beam with a wall filled with aluminum honeycomb and aluminum foam was established to predict the ultimate load. Then, an analytical model of a composite sandwich laminated box beam was established based on the Timoshenko beam theory. The analytical formulas of the overall stiffnesses of the composite sandwich laminated box beams were derived from internal force–displacement equations. Finally, the analytical model and the finite element model were used to study the effects of the ratio of circumferential layers to longitudinal layers and the ply angle on the ultimate load and on the maximum deflection at the mid-span. The effects of aluminum foam density and aluminum honeycomb cell wall thickness on the ratio of the overall stiffness to mass and the ratio of ultimate load to mass were also studied.

## 2. Experimental Study

### 2.1. Introduction of Specimens and Experimental Methods

In this paper, composite sandwich laminated box beams with cores of aluminum honeycomb, aluminum foam, and polyurethane elastomer were designed. The cross-sections of the specimens are shown in Figure 1. The inner and outer laminates of the sandwich panel were made of T300/QY8911 grade carbon-fiber composite material; the order of the layup was [0_4_/90]_s_, the thickness of the prepreg single layer was 0.15 mm, and the length of all three specimens, *l*, was 200 mm. The cross-sectional geometric dimensions are shown in Figure 2. The composite specimens adopted an overall secondary solidification molding process. The core material was bonded to the prepreg placed on the steel inner layer by layer, and then the first solidification was performed by vacuum bag pressing. After the first molding, the prepreg was rolled on the outside of the core material, and then subjected to secondary solidification. The composite sandwich laminated box beam specimens had better integrity and higher geometric dimensional accuracy. The specimen numbers and parameters are shown in Table 1. 

### 2.2. Three-Point Bending Experiment Process and Results

The three-point bending experiment was carried out on an INSTRON 5982 universal experimenting machine (Instron (Shanghai) Testing Equipment Trading Co., Ltd., Shanghai, China) with a load range of 100 kN. The strain gauge was attached to the bottom of the mid-span, and the strain data were recorded by a DH8303 strain acquisition analyzer. (Jiangsu Donghua Testing Technology Co., Ltd., Jingjiang City, Jiangsu Province, China). The whole experiment process was recorded by a digital camera (Canon (China) Co., Ltd., Beijing, China), as shown in Figure 3.

Figure 4 shows the three-point bending experiment process of the three composite sandwich laminated box beam beams. Figure 5 shows the load–displacement curves recorded by the experiment machine. It can be found that, at the beginning of the experiment, the specimen was bent and deformed, and the load increased with the displacement. When the ultimate load was reached, there was an auditory cracking sound. The upper flanges of the three composite sandwich laminated box beams all recessed downwards, indicating that nonlinear strength failure occurred in the action area of the indenter, and the structural bearing capacity started to fall. As the displacement of the experimenting machine increased, the breaking sound became more and more sharp and dense. Cracks appeared at the intersection of the webs and the upper flanges of the three composite sandwich laminated box beams, and out-of-plane bulging deformation occurred. For the composite polyurethane elastomer sandwich laminated box beam, the crack gradually expanded in the longitudinal direction, local buckling instability occurred on both webs, and the structural bearing capacity tended to be stable. For the composite aluminum honeycomb and the aluminum foam sandwich laminated box beams, vertical cracks were generated on the sides of the webs, and the cracks gradually expanded to the lower side of the web as the load increased. In general, the composite laminated box beams with the cores of aluminum honeycomb and aluminum foam had a small decrease in bearing capacity after strength failure, and the bearing capacity was stable within a certain range.

Figure 6 shows the load–mid-span strain curves of the composite sandwich laminated box beams in the linear elastic stage, and the overall bending stiffness *K_b_* of the composite sandwich laminated box beams could be calculated using the following formula [6,7]:(1)$$K_{b}=\frac{PLH}{8\varepsilon },$$where *P* is the concentrated load, *L* is the distance between the two seats, and *ε* is the mid-span strain.

According to the data of Figure 5 and Figure 6, the *K_b_* of the three kinds of specimens was calculated using Equation (1). In order to measure the benefits of light weight, the ratio of the ultimate load *P_u_* to the total mass *m* was defined as *P_mu_*, and the ratio of the overall bending stiffness *K_b_* to the total mass *m* was defined as *K_mb_*; the results are shown in Table 2. It can be found that the *P_u_* and *K_b_* of the composite aluminum foam sandwich laminated box beam was largest, while the *P_mu_* and *K_mb_* of the composite aluminum honeycomb sandwich laminated box beam was largest.

## 3. Model of Composite Sandwich Laminated Box Beam 

### 3.1. Finite Element Model of Composite Sandwich Laminated Box Beam

#### 3.1.1. Constitutive Model and Progressive Damage Analysis for Composite Panels

In Figure 7, (1, 2, 3) and (*n*, *s*, *z*) are the main axis and the off-axis coordinate systems of the orthotropic composite material, respectively, whereas *θ* is the angle between 1 and the *z*-direction. 

When *θ* = 0°, the three-dimensional constitutive equation of the composite is as follows:(2)$$\left\{\begin{array}{c} \sigma_{1}\\ \sigma_{2}\\ \sigma_{3}\\ \tau_{23}\\ \tau_{13}\\ \tau_{12} \end{array}\right\}=\left[\begin{array}{cccccc} C_{11} & C_{12} & C_{13} & 0 & 0 & C_{16}\\ C_{12} & C_{22} & C_{23} & 0 & 0 & C_{26}\\ C_{13} & C_{23} & C_{33} & 0 & 0 & C_{36}\\ 0 & 0 & 0 & C_{44} & C_{45} & 0\\ 0 & 0 & 0 & C_{45} & C_{55} & 0\\ C_{16} & C_{26} & C_{36} & 0 & 0 & C_{66} \end{array}\right]\left\{\begin{array}{c} \varepsilon_{1}\\ \varepsilon_{2}\\ \varepsilon_{3}\\ \gamma_{23}\\ \gamma_{13}\\ \gamma_{12} \end{array}\right\}$$

Equation (2) can be written as
***σ*** = ***Cε***,(3)
where ***C*** is the stiffness matrix of the main axis coordinate system, θ ≠ 0°, and the stiffness matrix of the off-axis coordinate system can be obtained as follows:(4)$$\sigma_{nsz}=TCT^{T}\varepsilon_{nsz}=\bar{C}\varepsilon_{nsz},$$where ***T*** is the transformation matrix, calculated as follows:(5)$$T=\left[\begin{array}{cccccc} \cos^{2}\theta & \sin^{2}\theta & 0 & 0 & 0 & -2\cos\theta \sin\theta \\ \sin^{2}\theta & \cos^{2}\theta & 0 & 0 & 0 & 2\cos\theta \sin\theta \\ 0 & 0 & 1 & 0 & 0 & 0\\ 0 & 0 & 0 & \cos\theta & \sin\theta & 0\\ 0 & 0 & 0 & -\sin\theta & \cos\theta & 0\\ \cos\theta \sin\theta & -\cos\theta \sin\theta & 0 & 0 & 0 & \cos^{2}\theta -\sin^{2}\theta \end{array}\right]$$

Substituting Equation (5) into Equation (4), the three-dimensional constitutive equation of the composite under the off-axis coordinate system is obtained by
(6)$$\left\{\begin{array}{c} \sigma_{z}\\ \sigma_{s}\\ \sigma_{n}\\ \tau_{sn}\\ \tau_{nz}\\ \tau_{sz} \end{array}\right\}=\left[\begin{array}{cccccc} \bar{C_{11}} & \bar{C_{12}} & \bar{C_{13}} & 0 & 0 & \bar{C_{16}}\\ \bar{C_{12}} & \bar{C_{22}} & \bar{C_{23}} & 0 & 0 & \bar{C_{26}}\\ \bar{C_{13}} & \bar{C_{23}} & \bar{C_{33}} & 0 & 0 & \bar{C_{36}}\\ 0 & 0 & 0 & \bar{C_{44}} & \bar{C_{45}} & 0\\ 0 & 0 & 0 & \bar{C_{45}} & \bar{C_{55}} & 0\\ \bar{C_{16}} & \bar{C_{26}} & \bar{C_{36}} & 0 & 0 & \bar{C_{66}} \end{array}\right]\left\{\begin{array}{c} \varepsilon_{z}\\ \varepsilon_{s}\\ \varepsilon_{n}\\ \gamma_{sn}\\ \gamma_{nz}\\ \gamma_{sz} \end{array}\right\},$$
where $\bar{C_{ij}}$ is the three-dimensional stiffness coefficient under the off-axis coordinate system. A more detailed expressions can be found in Reference [7].

The Hashin criterion was used as the damage initiation criterion of the composite. The Hashin criterion is a model correlation criterion, which can distinguish the four failure modes of fiber tensile, fiber crush, matrix tensile, and matrix crush, defined below.

(1) Fiber tensile failure
(7)$$F_{ft}={\left(\frac{\sigma_{11}}{X_{T}}\right)}^{2}+\alpha {\left(\frac{\tau_{12}}{S_{L}}\right)}^{2},\sigma_{11}\geq 0,$$

(2) Fiber crush failure
(8)$$F_{fc}={\left(\frac{\sigma_{11}}{X_{C}}\right)}^{2},\sigma_{11}\leq 0,$$

(3) Matrix tensile failure
(9)$$F_{mt}={\left(\frac{\sigma_{22}}{Y_{T}}\right)}^{2}+\alpha {\left(\frac{\tau_{12}}{S_{L}}\right)}^{2},\sigma_{22}\geq 0,$$

(4) Matrix crush failure
(10)$$F_{mc}={\left(\frac{\sigma_{22}}{2S_{T}}\right)}^{2}+\left[{\left(\frac{Y_{C}}{2S_{T}}\right)}^{2}-1\right]\frac{\sigma_{22}}{Y_{C}}+{\left(\frac{\tau_{12}}{S_{L}}\right)}^{2},\sigma_{22}\geq 0,$$
where *F_i_* (*i* = *ft*, *fc*, *mt*, *mc*) are the damage parameters corresponding to the four failure modes. When *F_i_* ≥ 1, the composite material is destroyed; *X*_T_ and *X*_C_ are the longitudinal tensile strength and compressive strength, respectively; *Y*_T_ and *Y*_C_ are the transverse tensile strength and compressive strength; *S*_L_ and *S*_T_ are the shear strength. The engineering elastic constants and ultimate strength values of the composite materials used in the panels are shown in Table 3.

When a single layer of the composite material is damaged in strength, its mechanical properties are attenuated to some extent. In this paper, the stiffness degradation was based on the linear degradation mode of fracture toughness, as shown in Figure 8, where σ_0,*i*_ (*i* = *ft*, *fc*, *mt*, *mc*) is the initial damage equivalent stress, *δ*_0,*i*_ is the initial damage displacement, *δ_f_*_,*i*_ is the complete damage displacement, and *d_i_* is the damage state variable. It can be seen from Figure 8 that the equivalent strain energy of the composite material under complete failure can be calculated as follows:(11)$$W_{s}=\frac{\sigma_{0.i}\delta_{f,i}}{2}$$

Table 4 shows the fracture energy parameters of the carbon-fiber composites used in the specimens [8]. When the composite material fails completely, the equivalent strain energy *W_s_* is equal to its fracture energy $G_{i}^{C}$. At this time, the complete damage displacement can be obtained as
(12)$$\delta_{f,i}=\frac{2G_{i}^{C}}{\sigma_{0,i}}$$

The degree of deterioration of the mechanical properties of the material is characterized by the damage state variable *d_i_*, and their expression is as follows:(13)$$d_{i}=\frac{\delta_{f,i}(\delta_{i}-\delta_{0,i})}{\delta_{i}(\delta_{f,i}-\delta_{0,i})}$$

The constitutive equation of composite monolayers in the process of damage evolution is as follows:***σ*** = **C*_d_ε***,(14)
where ***C_d_*** is the stiffness matrix considering the damage, calculated as
(15)$$C_{d}=\frac{1}{D}\left[\begin{array}{ccc} (1-d_{f})E_{1} & (1-d_{f})(1-d_{m})v_{21}E_{1} & 0\\ (1-d_{f})(1-d_{m})v_{21}E_{1} & (1-d_{m})v_{21}E_{2} & 0\\ 0 & 0 & D(1-d_{s})G_{12} \end{array}\right]$$
where $D=1-(1-d_{f})(1-d_{m})$; *d_f_*, *d_m_*, and *d_s_* are state variables of fiber damage, matrix damage, and in-plane shear damage, respectively, which can be expressed as follows
(16)$$d_{f}=\{\begin{array}{c} d_{ft}\begin{array}{cc} & \sigma_{11}\geq 0 \end{array}\\ d_{fc}\begin{array}{cc} & \sigma_{11}<0 \end{array} \end{array},$$
(17)$$d_{m}=\{\begin{array}{c} d_{mt}\begin{array}{cc} & \sigma_{11}\geq 0 \end{array}\\ d_{mc}\begin{array}{cc} & \sigma_{11}<0 \end{array} \end{array},$$
(18)$$d_{s}=1-(1-d_{ft})(1-d_{fc})(1-d_{mt})(1-d_{mc})$$

The relationship between the effective stress matrix and the real stress matrix is as follows:(19)$$\bar{\sigma }=M\sigma$$

***M*** is the damage coefficient matrix and is calculated by
(20)$$M=\left[\begin{array}{ccc} \frac{1}{1-d_{f}} & 0 & 0\\ 0 & \frac{1}{1-d_{m}} & 0\\ 0 & 0 & \frac{1}{1-d_{s}} \end{array}\right]$$

#### 3.1.2. Elastoplastic Constitutive Model of Core Material

For aluminum honeycomb and aluminum foam core materials, an ideal elastoplastic model was used. The aluminum honeycomb core substrate used in this paper was pure aluminum with a modulus of elasticity of 67 GPa, a yield strength of 108 MPa, and a Poisson’s ratio of 0.3. The stress–strain relationship of pure aluminum is shown in Figure 9 [9]. The density of the aluminum foam core material was 0.5 g/cm^3^, and the aluminum foam constitutive model considering the effect of density can be represented by Equation (21) [10].
(21)$$\sigma =20.34\rho^{1.69}\frac{e^{94.58\rho^{0.08}\varepsilon }-1}{1+e^{94.41\rho^{0.08}\varepsilon }}+e^{12.34\rho -10.55}\left[e^{\left(-8.6\rho +14.68\right)\varepsilon }-1\right]$$

The stress–strain curves of aluminum foam with different densities are shown in Figure 10.

#### 3.1.3. Adhesive Interface Element Damage Analysis Model

The interface performance was simulated by placing interface elements of COH3D8. COH3D8 is a three-dimensional eight-node interface element with thickness as shown in Figure 11 [11].

The constitutive equation of COH3D8 in the linear elastic range is as follows [11]:(22)$$t=\left\{\begin{array}{c} t_{n}\\ t_{s}\\ t_{t} \end{array}\right\}\left[\begin{array}{ccc} K_{nn} & 0 & 0\\ 0 & K_{ss} & 0\\ 0 & 0 & K_{tt} \end{array}\right]\left\{\begin{array}{c} \varepsilon_{n}\\ \varepsilon_{s}\\ \varepsilon_{t} \end{array}\right\}=K\varepsilon$$where *t_n_*, *t_s_*, and *t_t_* are the normal stress and the two shear stresses of the adhesive interface element respectively, *ε_i_* = *δ_i_*/*T*_0_ (*i* = *n*, *s*, *t*), *T*_0_ is the thickness actually calculated for the adhesive element, *δ_i_* is the relative displacement of the top surface and the bottom surface of the cohesive element in the corresponding direction, and *K_ii_* is a stiffness factor. The secondary stress criterion was used as the starting criterion, shown in Equation (23).
(23)$${\left(\frac{t_{n}}{t_{n}^{0}}\right)}^{2}+{\left(\frac{t_{s}}{t_{s}^{0}}\right)}^{2}+{\left(\frac{t_{t}}{t_{t}^{0}}\right)}^{2}=1$$

The damage evolution of COH3D8 elements was based on the mixed mode Benzeggagh–Kenane energy criterion [12], shown below.
(24)$$G_{n}^{C}+(G_{s}^{C}-G_{n}^{C}){\left(\frac{G_{s}+G_{t}}{G_{n}+G_{s}+G_{t}}\right)}^{\eta }=G^{C},$$
where *G_i_* (*i* = *n*, *s*, *t*) represents the strain energy release rates corresponding to open, slip, and tear cracks, respectively, $G_{i}^{C}$ represents the critical strain energy release rates of the three kinds of cracks, $G^{C}$ is the damage evolution variable (the layers are fully stratified when $G^{C}=1$), and *η* is the damage factor. For carbon-fiber epoxy materials, *η* is generally between 1 and 2. The material parameters of the adhesive element layer are shown in Table 5 [13].

#### 3.1.4. Geometric Model

The inner and outer panels were cut into 10 layers in the thickness direction. Each layer was simulated by an SC8R continuous shell element with a thickness of 0.15 mm. The “composite layup” function in ABAQUS was used to grant material properties to the inner and outer panels. The S4R common shell element and the C3D8R three-dimensional stress element were used to simulate the aluminum honeycomb and the aluminum foam core material. A layer of COH3D8 interface element with zero thickness was arranged between the core material and the inner and the outer panels to simulate degumming damage. A universal contact with a tangential friction coefficient of 0.1 and a normal “hard” contact [14] was established. A 38 mm × 10 mm node area was created in the middle of the upper flange to establish a coupling constraint between the node area and the reference point RP1. The displacement load on RP1 was applied, and a historical output of U2 and RF2 was established. The geometric model of the two supports was also established and defined as a rigid body; then, a fully fixed constraint was applied. The finite element models of the two composite sandwich laminated box beams are shown in Figure 12.

### 3.2. Analytical Model of Composite Sandwich Laminated Box Beam

#### 3.2.1. Simplification of Constitutive Equation for Composite Panels and Core Materials

In order to describe the deformation geometry of the composite sandwich laminated box beams, the Cartesian coordinate system (*x*, *y*, *z*) was used for the overall coordinates, whereby the coordinate origin was located in the cross-section centroid; the curvilinear coordinate system (*z*, *s*, *n*) was used for the local coordinates, whereby the origin of the coordinates was in the midline of the cross-section; *n* and *s* represent the normal direction and the tangential direction of the middle line, respectively. The core material and the panels were all symmetric about the middle line, as shown in Figure 13.

The inner and outer panels were composite laminate structures, and the three-dimensional constitutive equation of an arbitrary *k*-th composite single layer is shown in Equation (6). The stress components *σ_n_*, *σ_s_*, and *τ_sn_* outside the cross-section were much smaller than the in-plane stress components *σ_z_*, *τ_nz_*, and *τ_sz_*, where it can be assumed that *σ_n_ = σ_s_ = τ_sn_* = 0.
(25)$$\{\begin{array}{l} \sigma_{n}=\bar{C_{31}}\varepsilon_{z}+\bar{C_{23}}\varepsilon_{s}+\bar{C_{33}}\varepsilon_{n}+\bar{C_{36}}\gamma_{sz}=0\\ \sigma_{s}=\bar{C_{21}}\varepsilon_{z}+\bar{C_{22}}\varepsilon_{s}+\bar{C_{23}}\varepsilon_{n}+\bar{C_{26}}\gamma_{sz}=0\\ \tau_{sn}=\bar{C_{44}}\gamma_{sn}+\bar{C_{45}}\varepsilon_{nz}=0 \end{array}$$

Equation (25) was substituted into Equation (6), which could be simplified as follows:(26)$$\left\{\begin{array}{c} \sigma_{z}\\ \tau_{sz}\\ \tau_{nz} \end{array}\right\}=\left[\begin{array}{ccc} C_{11}^{*} & C_{12}^{*} & 0\\ C_{12}^{*} & C_{22}^{*} & 0\\ 0 & 0 & C_{33}^{*} \end{array}\right]\left\{\begin{array}{c} \varepsilon_{z}\\ \gamma_{sz}\\ \gamma_{nz} \end{array}\right\}$$

The formulas for calculating the three-dimensional converted stiffness coefficient $C_{ij}^{*}$ in Equation (26) were as follows:(27)$$\begin{array}{l} C_{11}^{*}=\bar{Q_{11}}-\frac{{\bar{Q_{12}}}^{2}}{\bar{Q_{22}}},\;C_{12}^{*}=\bar{Q_{16}}-\frac{\bar{Q_{12}}\bar{Q_{26}}}{\bar{Q_{22}}},\\ C_{22}^{*}=\bar{Q_{66}}-\frac{{\bar{Q_{26}}}^{2}}{\bar{Q_{22}}},\;C_{33}^{*}=\bar{Q_{55}}-\frac{{\bar{Q_{45}}}^{2}}{\bar{Q_{44}}} \end{array}$$where $\bar{Q_{ij}}$ is the two-dimensional converted modulus component in the classical laminate theory.

The aluminum honeycomb core material is also an orthotropic material, but it does not have a change in ply angle like the single layer of the composite material. The main axis coordinate system (1, 2, 3) and the local coordinate system (*s*, *n*, *z*) of aluminum honeycomb were parallel, reducing Equation (27) to
(28)$$C_{11}^{*}=\frac{E_{1}}{1-v_{12}^{2}}(1-\frac{v_{12}^{2}E_{1}}{E_{2}}),\;C_{12}^{*}=0,\;C_{22}^{*}=\;G_{12},\;C_{33}^{*}=G_{13}$$

*E*_1_, *E*_2_, *G*_1__2_, and *G*_1__3_ in Equation (28) are the equivalent elastic parameters of aluminum honeycomb, the calculation method of which is detailed in Reference [15]. It can be found that the aluminum honeycomb core material has no tensile–shear coupling effect.

For isotropic materials such as aluminum foam and polyurethane elastomers, Equation (28) can be further simplified as
(29)$$C_{11}^{*}=E,\;C_{12}^{*}=0,\;C_{22}^{*}=C_{33}^{*}=G$$

It can be seen that, whether it is a laminated wall or a core material, the constitutive equation can be represented by Equation (26). The orthotropic, isotropic core material can be equivalent to a single layer of composite material. There is no tensile–shear coupling effect in the constitutive equation of isotropic materials. Subsequent calculations can be simplified by equating the sandwich panels to common laminates.

#### 3.2.2. Calculation of Bending Stiffness of Composite Sandwich Laminated Box Beam

It was assumed that the panel, the core material, and the neighboring layers were all firmly bonded, and no relative slip occurred between them. By integrating the normal stress *σ*_z_ and the two shear stresses *τ*_sz_ and *τ*_nz_ along the section, the internal force and the internal moment could be obtained as follows [16]:(30)$$\{\begin{array}{l} N_{z}=\int_{A}\sigma_{z}dsdn\\ V_{x}=\int_{A}(\tau_{sz}\cos\phi +\tau_{nz}\sin\phi )dsdn\\ V_{y}=\int_{A}(\tau_{sz}\sin\phi -\tau_{nz}\cos\phi )dsdn\\ M_{x}=\int_{A}\sigma_{z}(y-n\cos\phi )dsdn\\ M_{y}=\int_{A}\sigma_{z}(x+n\sin\phi )dsdn\\ M_{z}=\int_{A}\tau_{sz}\psi (s)dsdn\\ M_{\omega }=\int_{A}\sigma_{z}[F_{w}(s)+na(s)]dsdn \end{array}$$where *N_z_* is the axial force, *V_x_* and *V_y_* are the shear forces in the *x* and *y* directions, respectively, *M_x_*, *M_y_*, and *M_z_* are the bending moments around the *x*, *y*, and *z* axes, respectively, *M_w_* is the bi-moment generated by the torsional normal stress, *ϕ* is the angle between the *n* and the *x* directions, *a*(*s*) is the height of the right triangle in the geometric relationship, *F_w_*(s) is the generalized fan coordinate, and *ψ*(s) is the warping function.

The relationship between internal force and displacement can be obtained by simplifying Equation (30) [16].
(31)$$\left\{\begin{array}{c} N_{z}\\ M_{y}\\ M_{x}\\ V_{x}\\ V_{y}\\ M_{\omega }\\ M_{z} \end{array}\right\}=\left[\begin{array}{ccccccc} a_{11} & a_{12} & a_{13} & a_{14} & a_{15} & a_{16} & a_{17}\\ & a_{22} & a_{23} & a_{24} & a_{25} & a_{26} & a_{27}\\ & & a_{33} & a_{34} & a_{35} & a_{36} & a_{37}\\ & & & a_{44} & a_{45} & a_{46} & a_{47}\\ & & & & a_{55} & a_{56} & a_{57}\\ & & & & & a_{66} & a_{67}\\ sym & & & & & & a_{77} \end{array}\right]\left\{\begin{array}{c} w_{0}^{\prime }\\ \theta_{y}^{\prime }\\ \theta_{x}^{\prime }\\ u_{0}^{\prime }+\theta_{y}\\ v_{0}^{\prime }+\theta_{x}\\ \varphi^{\prime \prime }\\ \varphi^{\prime } \end{array}\right\},$$
where *u*, *v*, and *w* are displacements along the coordinate axes *x*, *y*, and *z*, respectively, *θ_x_*(*z*), *θ_x_*(*z*), and *φ*(*z*) are rotation angles around the coordinate axes *x*, *y*, and *z* respectively, and *u*_0_(*z*), *v*_0_(*z*), and *w_0_*(*z*) are rigid body displacements in the three directions. The direction of the displacement components and the internal force components in Equation (31) are shown in Figure 14.

When the laminated panel adopts balanced oblique symmetric, balanced antisymmetric, and orthogonal layups, *a_ij_* = 0 (*i* ≠ *j*), the elastic coupling effects between different deformations are all cancelled [16,17]; thus, Equation (31) can be reduced to
(32)$$\begin{array}{l} w_{0}^{\prime }=\frac{N_{z}}{a_{11}},\;\theta_{y}^{\prime }=\frac{M_{y}}{a_{22}},\;\theta_{x}^{\prime }=\frac{M_{x}}{a_{33}},\;u_{0}^{\prime }+\theta_{y}=\frac{V_{x}}{a_{44}}\\ v_{0}^{\prime }+\theta_{x}=\frac{V_{y}}{a_{55}},\;\phi^{\prime \prime }=\frac{M_{\omega }}{a_{66}},\;\phi^{\prime }=\frac{M_{z}}{a_{77}} \end{array}$$

From the relationship between internal force and displacement in Equation (32), the stiffnesses of the composite laminated box beams can be obtained: axial stiffness (*EA*); bending stiffness (*EI*)*_y_*, (*EI*)*_x_* for the *y* and *x* axes; shear stiffness (*GA*)*_y_*, (*GA*)*_x_* for the *y* and *x* axes; the section warping stiffness (*GI*)*_w_*; the section torsional stiffness (*GJ*). The calculation formulas of the stiffnesses mentioned above are as follows:(33)$$\begin{array}{l} \left[EA\right]=a_{11}=\int_{s}\left(A_{11}-\frac{A_{12}^{2}}{A_{22}}\right)ds\\ {\left[EI\right]}_{y}=a_{22}=\int_{s}\left[x^{2}(A_{11}-\frac{A_{12}^{2}}{A_{22}})+2(B_{11}-\frac{A_{12}B_{12}}{A_{22}})x\sin\theta +(D_{11}-\frac{B_{12}^{2}}{A_{22}})\sin^{2}\theta \right]ds\\ {\left[EI\right]}_{x}=a_{33}=\int_{s}\left[y^{2}(A_{11}-\frac{A_{12}^{2}}{A_{22}})-2y(B_{11}-\frac{A_{12}B_{12}}{A_{22}})\cos\theta +(D_{11}-\frac{B_{12}^{2}}{A_{22}})\cos^{2}\theta \right]ds\\ {\left[GA\right]}_{x}=a_{44}=\int_{s}\left[\left(A_{66}-\frac{A_{26}^{2}}{A_{22}}\right)\cos^{2}\theta +\left(A_{55}-\frac{A_{45}^{2}}{A_{44}}\right)\sin^{2}\theta \right]ds\\ {\left[GA\right]}_{y}=a_{55}=\int_{s}\left[\left(A_{66}-\frac{A_{26}^{2}}{A_{22}}\right)\sin^{2}\theta +\left(A_{55}-\frac{A_{45}^{2}}{A_{44}}\right)\cos^{2}\theta \right]ds\\ {\left[EI\right]}_{w}=a_{66}=\int_{s}\left(A_{11}F_{\omega }^{2}+2B_{11}F_{\omega }a+D_{11}a^{2}\right)ds\\ \left[GJ\right]=a_{77}=\int_{s}\left(A_{66}-\frac{A_{26}^{2}}{A_{22}}\right)\psi^{2}(s)ds \end{array}$$where *A_ij_*, *B_ij_*, and *D_ij_* are the in-plane stiffness coefficients, coupling stiffness coefficients, and bending stiffness coefficients in the classical laminate theory.

### 3.3. Results, Discussion, and Comparative Analysis

#### 3.3.1. Model Validity Verification

The bending stiffnesses of the composite sandwich laminated box beams were calculated by Equation (33) and compared with the experiment results, as shown in Table 6. It can be found that the errors calculated by Equation (33) were within 6%, meeting the accuracy requirement of engineering applications.

The load–displacement curves of the three-point bending process of the composite sandwich laminated box beam obtained by the experiment and the finite element simulation are shown in Figure 15. The overall change trend of the load–displacement curves obtained by the two methods was similar. The numerical ultimate loads of the two different composite sandwich laminated box beams were 7.59 kN and 11.2 kN. Compared with the experimental data in Table 2, the errors were 6.7% and 6.9%, respectively. The reliability of the finite element model is, therefore, illustrated.

#### 3.3.2. Failure Mechanism Analysis of Composite Sandwich Laminated Box Beams under Three-Point Bending

Figure 16 shows the results of the finite element simulation; there were four failure modes of the composite sandwich laminated box beams under ultimate load. According to the legend in Figure 16, the scale from blue to red indicates that the damage is getting worse. Composite damages occurred in areas of the circle, where the red area indicates that elements were completely destroyed. It can be seen that, when the ultimate load was reached, the compression zone of the two composite sandwiches began to undergo partial depression deformation, where the stress concentration at the top of the web was the most serious. The collapse of the fiber and matrix in the compression zone of the upper flange and the top of web was the main cause of the decline in bearing capacity.

Figure 17 shows the stress cloud of the composite sandwich box beam and core material after nonlinear deformation. A darker color denotes a more severe stress concentration. After the ultimate load, as the displacement increased, the web near the compression zone of the outer panel underwent out-of-plane bulging deformation, part of the failure element was removed, and the lower flange was mainly subjected to tensile stress. Both the aluminum honeycomb and the aluminum foam core materials underwent overall bending deformation and local plastic deformation. The panels and the core material were degummed and dislocated, and the deformation of the inner and outer panels was uncoordinated.

## 4. Analysis of Factors Affecting Mechanical Properties of Composite Sandwich Laminated Box Beams

For composite sandwich laminated box beams subjected to concentrated loads at the mid-span, the maximum mid-span deflection caused by the bending load [3] was calculated as follows:(34)$$\delta_{M}=\frac{PL^{3}}{48{\left[EI\right]}_{x}}$$

The additional deflection at the mid-span caused by the transverse shear force was calculated as follows:(35)$$\delta_{V}=\alpha \frac{PL}{4{\left[GA\right]}_{y}},$$where *P* is the concentrated load, *L* is the distance between the two supports, *α* is the shear section coefficient, and [*EI*]*_x_*, [*GA*]*_y_* are the bending stiffness and shear stiffness calculated by Equation (33).

The total deflection *δ* of the span was calculated as follows:(36)$$\delta =\delta_{M}+\delta_{V}$$

Under the premise of considering lateral shear deformation and with the purpose of characterizing the overall stiffness performance of the composite sandwich laminated box beam and measuring the benefits of light weight, the overall stiffness coefficient *K* and the specific stiffness coefficient *K_m_* were defined as follows:(37)$$K=\frac{1}{\delta },$$
(38)$$K_{{}_{m}}=\frac{1}{m\delta }$$

### 4.1. Influence of the Ratio of Span to Height 

The composite aluminum honeycomb sandwich laminated box beam in Section 2.1 was taken as an example. It was assumed that the cross-section dimensions of the laminated box beam and the inner and outer panel layups were unchanged, and the concentrated load was *P* = 1000 N. Using the laminated box beam span *L* as a variable, *δ_M_*, *δ_V_*, and *δ* were calculated by Equations (34)–(36), respectively. The results are shown in Figure 18. It can be seen that the ratio of the shear additional deflection *δ_V_* to the total deflection *δ* decreased as the ratio of span to height *L*/*H* increased; when *L*/*H* was less than 10, the ratio of *δ_V_* to *δ* was above 25%, and, for the isotropic material laminated box beam, the ratio of *δ_V_* to *δ* was only 3%. The composite sandwiched box beam was more affected by shear deformation.

### 4.2. Influence of Layup Parameters

The composite aluminum honeycomb core (AHC) and aluminum foam core (AFC) laminated box beams in Section 2.1 were taken as examples. It was assumed that the inner and outer panels only adopted the layers of 0 degrees and 90 degrees, indicating that the layers of the composite sandwich laminated box beams were only in the longitudinal and circumferential directions. The core material was unchanged. The finite element model and the analytical model were used to calculate the ultimate load and stiffness coefficients of the composite laminated box beams with different ratios of circumferential layers to longitudinal layers, and the results are shown in Figure 19. It can be found that the composite sandwich laminated box beam with AFC had relatively large ultimate load and stiffness coefficients, which was due to the higher equivalent elastic modulus and yield strength of the aluminum foam. When the panel had a pure 0-degree or a pure 90-degree layup, the ultimate load was lower. Furthermore, the stiffness coefficient of the composite sandwich laminated box beam decreased as the ratio of circumferential layers to longitudinal layers increased. This is because the 90-degree layers increased the lateral load-bearing capacity and the transverse stiffness of the web, but the small number of 0-degree layers reduced the overall bending stiffness of the composite sandwich box beam.

The composite aluminum honeycomb sandwich laminated box beam was taken as an example. It was assumed that the inner and outer wall both had a balanced obliquely symmetric layup, [*θ*/−*θ*]_5_; the variations of [*EI*]*_x_*, [*GA*], *δ_M_*, *δ_V_*, and *δ* changing with the ply angle *θ* are shown in Figure 20 and Figure 21. It can be found that [*EI*]*_x_*, [*GA*], and *K* took maximum values at 0, 45, and 15 degrees, respectively.

### 4.3. Influence of Core Material Parameters

It was assumed that the cross-sectional geometry, and the inner and outer panel layups of the composite sandwich laminated box beam were unchanged, and the concentrated load was *P* = 1000 N. The mechanical properties of the composite sandwich laminated box beam were analyzed with different densities of aluminum foam and different wall thicknesses of the aluminum honeycomb cell. The results are shown in Figure 22 and Figure 23. It can be found that, as the density of aluminum foam and the wall thickness of the aluminum honeycomb cell increased, the values of *K_m_* and *P_um_* decreased, and the benefits of light weight decreased.

## 5. Conclusions

(1)Composite sandwich laminated box beams with cores of aluminum honeycomb, aluminum foam, and polyurethane elastomer underwent localized crushing damage under three-point bending load. After the ultimate load, the first two kinds of composite sandwich laminated box beams still had a high load-carrying capacity within a certain range. The composite aluminum honeycomb sandwich box laminated beam had the highest ratios of overall stiffness to mass and ultimate load to mass, leading to a better benefit of light weight.(2)The finite element models of composite sandwich laminated box beams filled with aluminum honeycomb and aluminum foam, established by composite progressive damage analysis and the core elastoplastic constitutive equation, could approximately simulate the three-point bending failure process and predict the ultimate load. The analytical model of the composite sandwich laminated box beam established by the Timoshenko beam theory could approximately calculate the overall stiffness parameters of the composite sandwich laminated box beam.(3)For composite sandwich laminated box beams with a small ratio of span to height, the additional deflection caused by shear deformation must be considered in the deflection analysis. As the ratio of circumferential layers to longitudinal layers increased, the three-point bending ultimate load of the composite sandwich laminated box beam increased, and the overall bending stiffness decreased. The use of aluminum foam with low density and aluminum honeycomb with a small cell-wall thickness as the core materials can achieve greater benefits of light weight.

## Figures and Tables

**Figure 1 materials-12-02959-f001:**
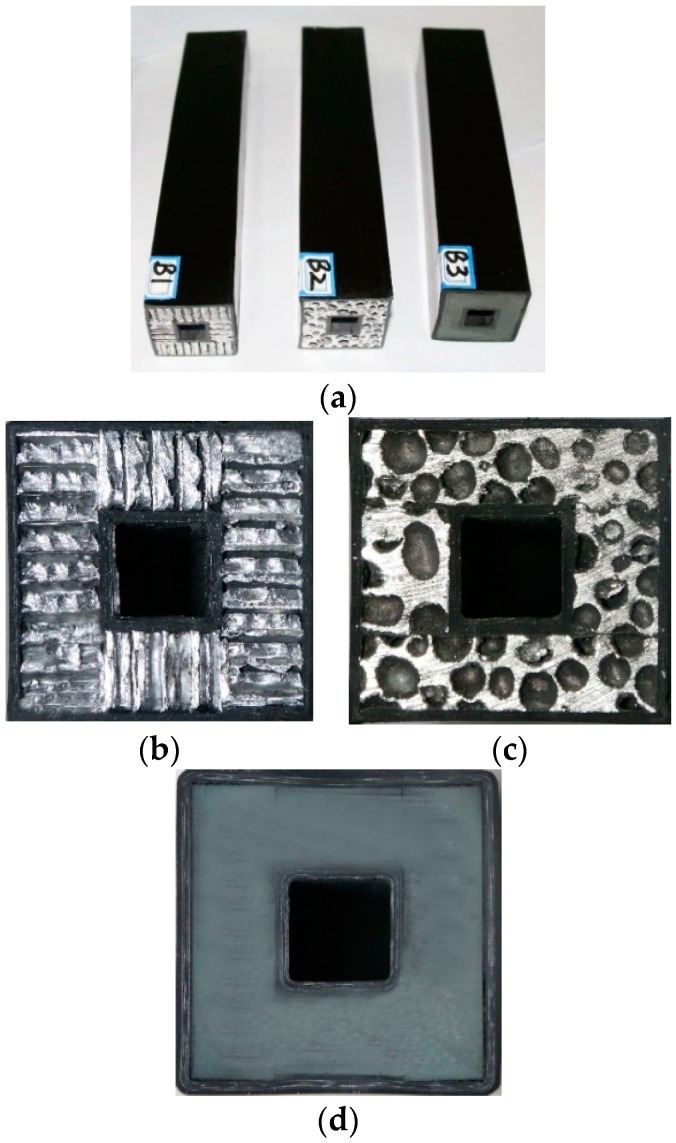
Composite sandwich laminated box beams with three different core materials: (**a**) composite sandwich laminated box beam specimens; (**b**) aluminum honeycomb core; (**c**) aluminum foam core; (**d**) polyurethane elastomer core.

**Figure 2 materials-12-02959-f002:**
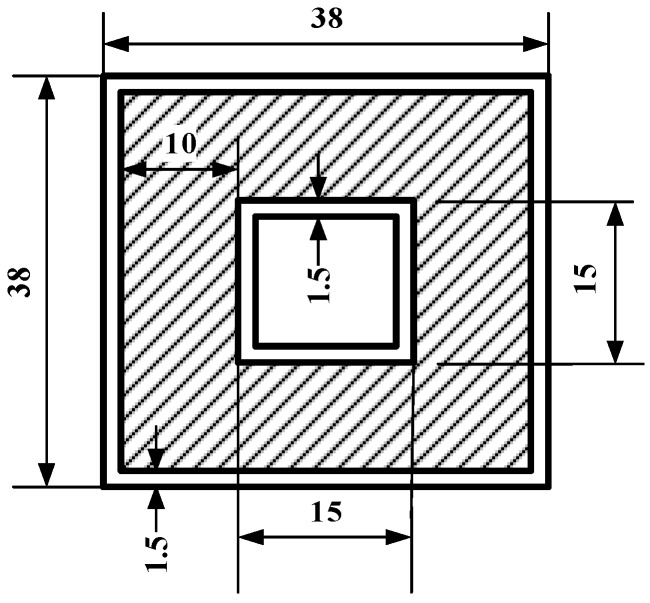
Geometric dimensions of composite sandwich laminated box beams.

**Figure 3 materials-12-02959-f003:**
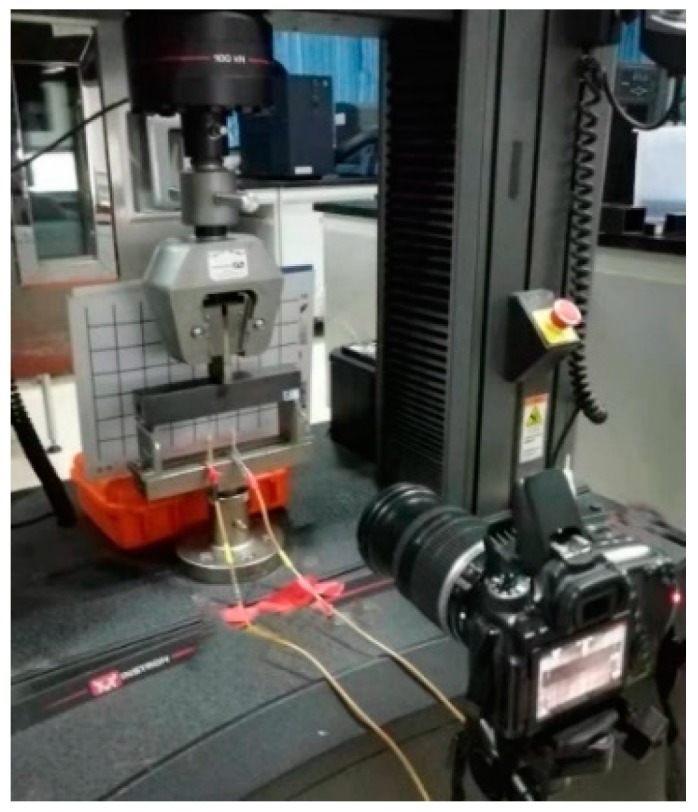
Schematic of the three-point bending experiment.

**Figure 4 materials-12-02959-f004:**
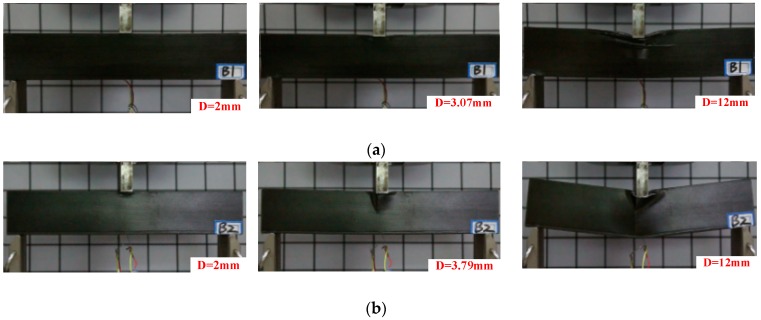
Three-point bending experiment process of composite sandwich laminated box beams. (**a**) Aluminum honeycomb sandwich; (**b**) Aluminum foam sandwich; (**c**) Polyurethane elastomer sandwich.

**Figure 5 materials-12-02959-f005:**
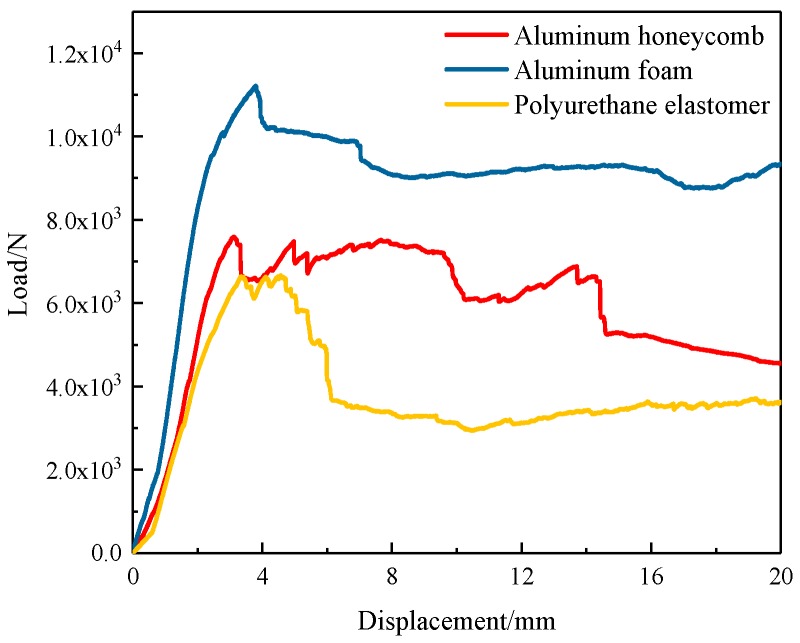
Load–displacement curves of the three composite sandwich laminated box beams.

**Figure 6 materials-12-02959-f006:**
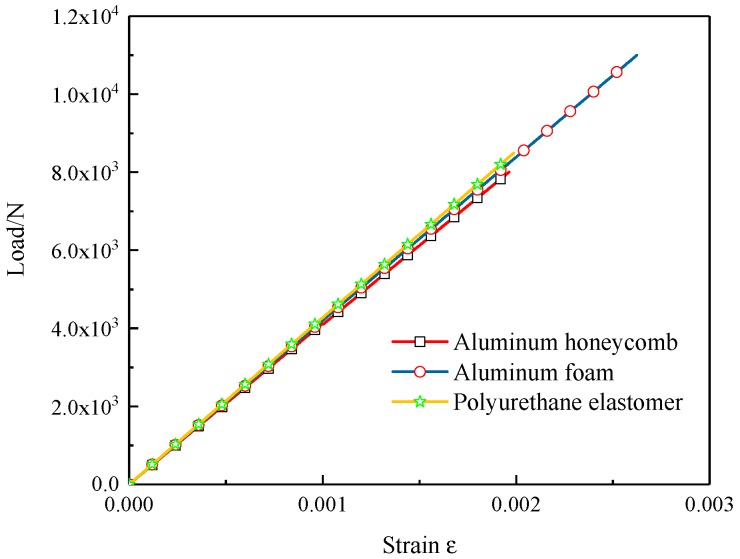
Load–strain curves of the three different composite sandwich laminated box beams.

**Figure 7 materials-12-02959-f007:**
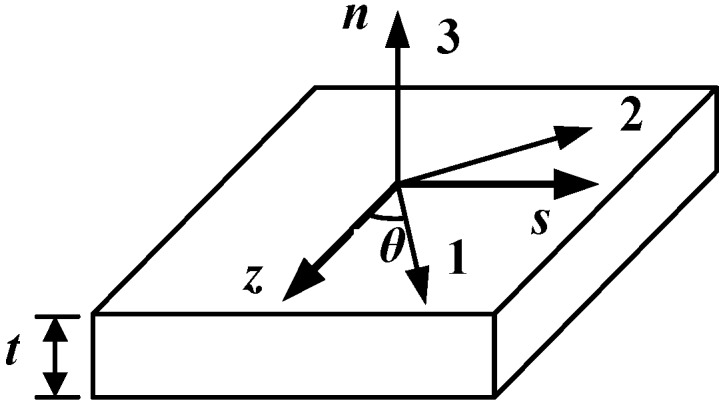
The coordinate system of the composite.

**Figure 8 materials-12-02959-f008:**
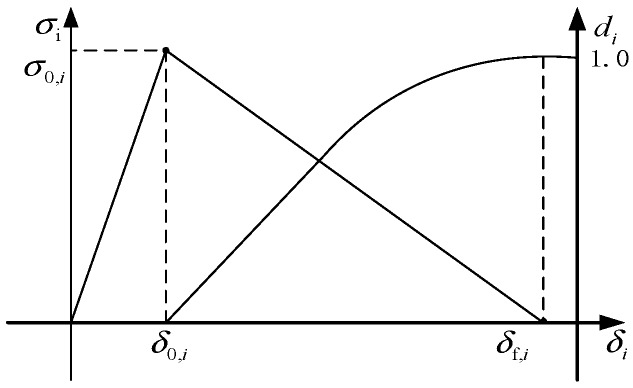
Linear degradation mode.

**Figure 9 materials-12-02959-f009:**
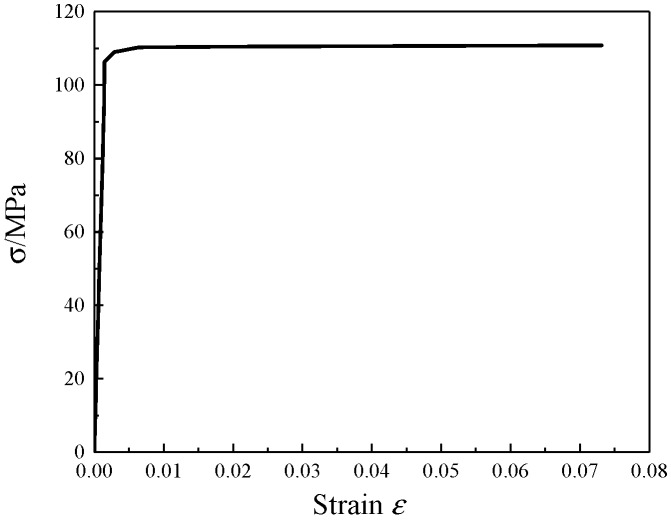
The stress–strain curve of the aluminum honeycomb matrix.

**Figure 10 materials-12-02959-f010:**
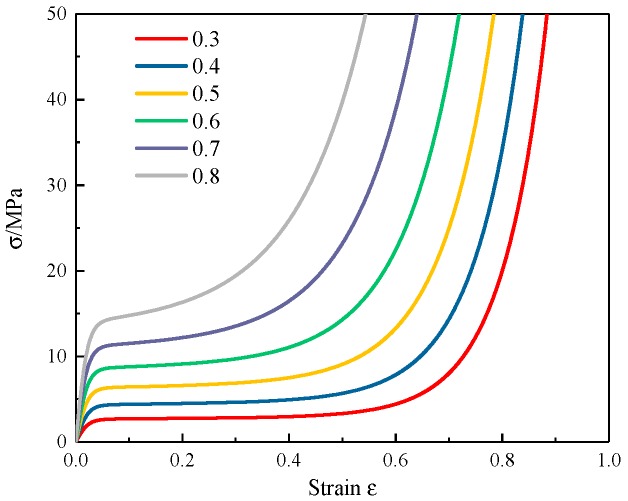
The stress–strain curves of aluminum foam with different densities.

**Figure 11 materials-12-02959-f011:**
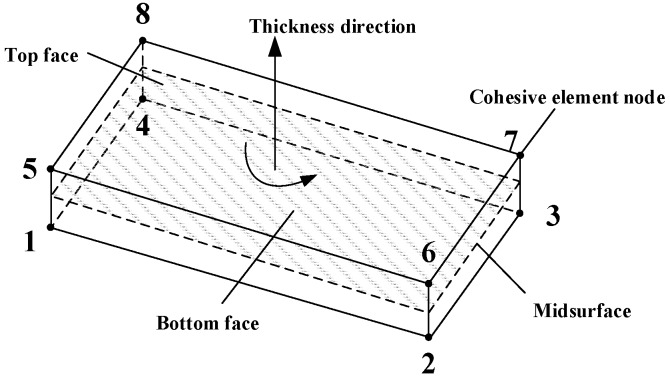
Schematic of three-dimensional eight-node interface element.

**Figure 12 materials-12-02959-f012:**
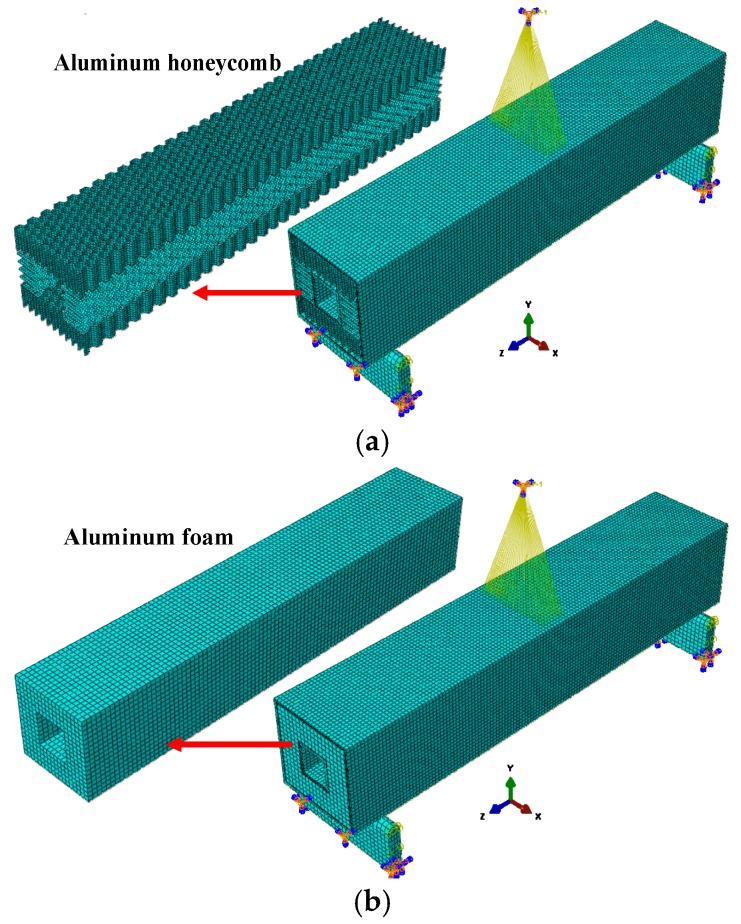
Finite element models of composite sandwich laminated box beams. (**a**) Aluminum honeycomb core; (**b**) Aluminum foam core.

**Figure 13 materials-12-02959-f013:**
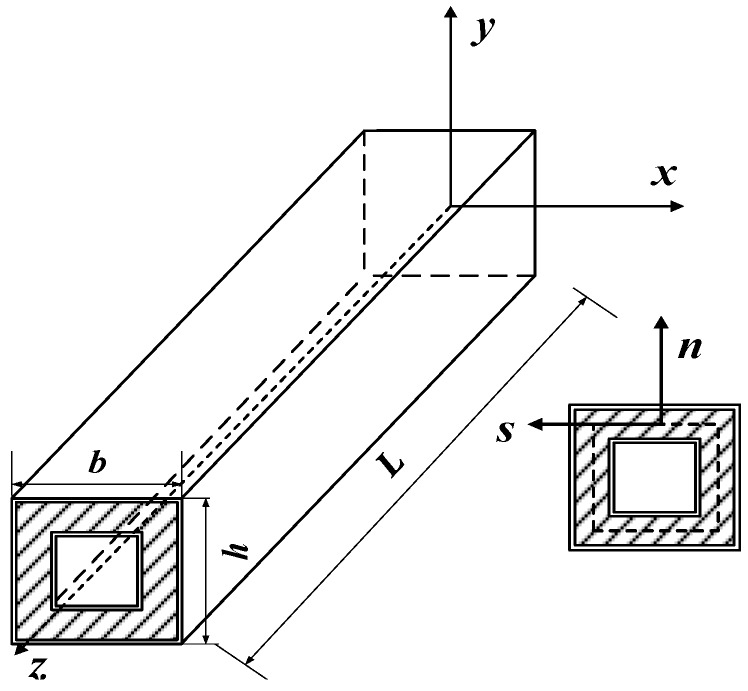
Composite sandwich laminated box beam coordinate system.

**Figure 14 materials-12-02959-f014:**
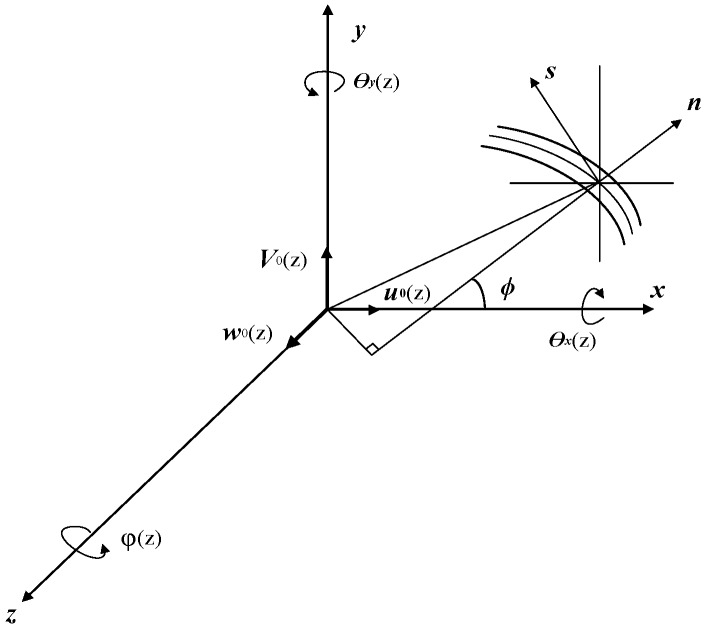
Internal force and displacement components of composite sandwich laminated box beam.

**Figure 15 materials-12-02959-f015:**
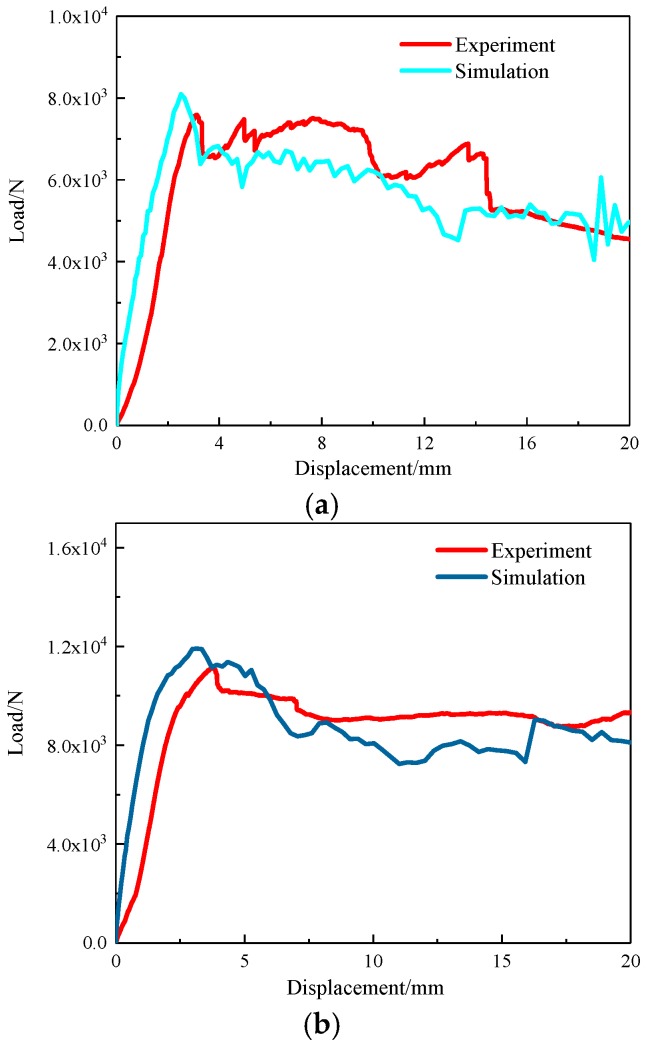
Load–displacement curves of three-point bending process of composite sandwich laminated box beams. (**a**) Aluminum honeycomb core; (**b**) Aluminum foam core.

**Figure 16 materials-12-02959-f016:**
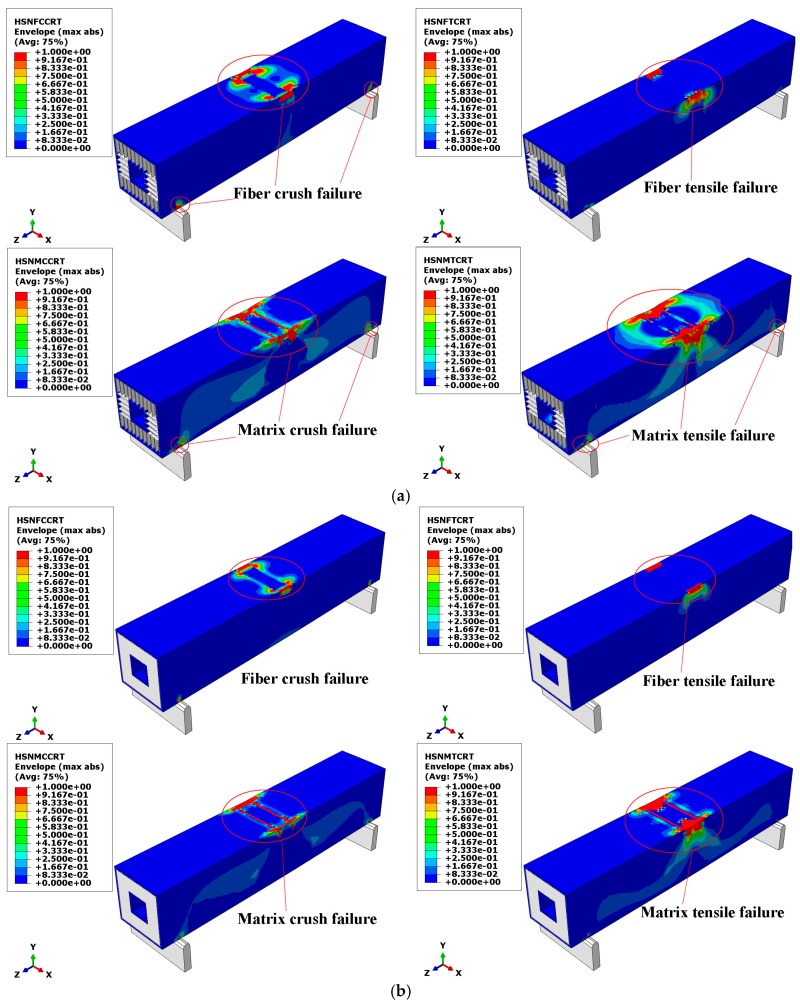
Failure modes of composite sandwich laminated box beam. (**a**) Aluminum honeycomb core; (**b**) Aluminum foam core.

**Figure 17 materials-12-02959-f017:**
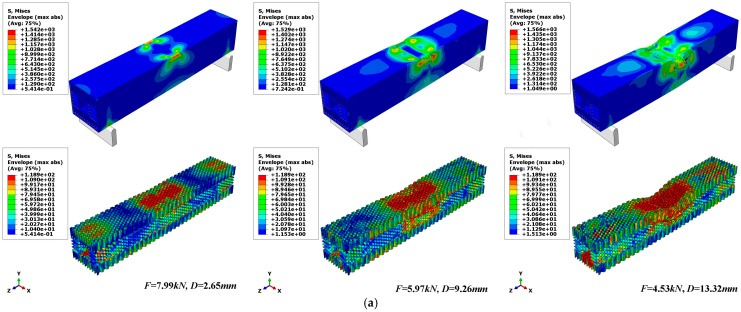
The stress clouds of the whole composite sandwich laminated box beams under three-point bending failure process. (**a**) Aluminum honeycomb core; (**b**) Aluminum foam core.

**Figure 18 materials-12-02959-f018:**
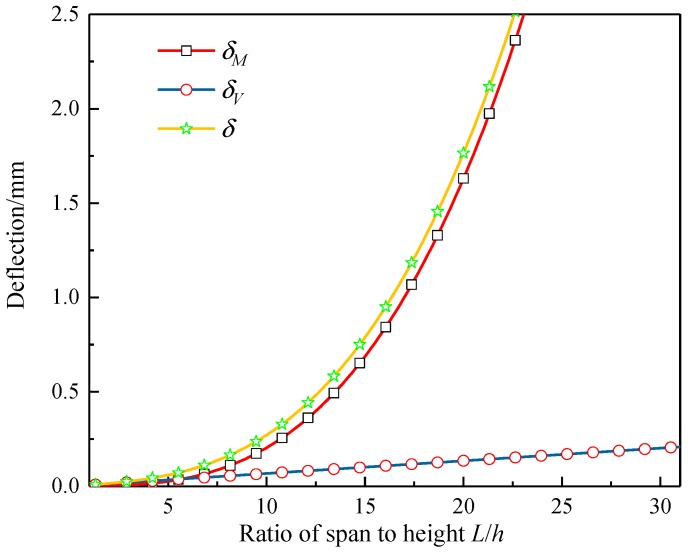
Curves of bending deflection *δ_M_*, shearing additional deflection *δ_V_*, and total deflection *δ* with the ratio of span to height *L*/*H.*

**Figure 19 materials-12-02959-f019:**
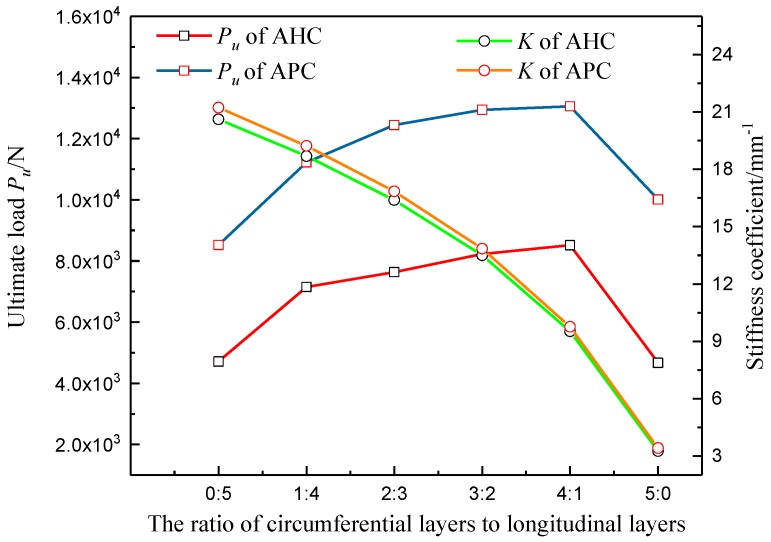
Curves of ultimate load *P_u_* and stiffness coefficient *K* changing with the ratio of circumferential layers to longitudinal layers.

**Figure 20 materials-12-02959-f020:**
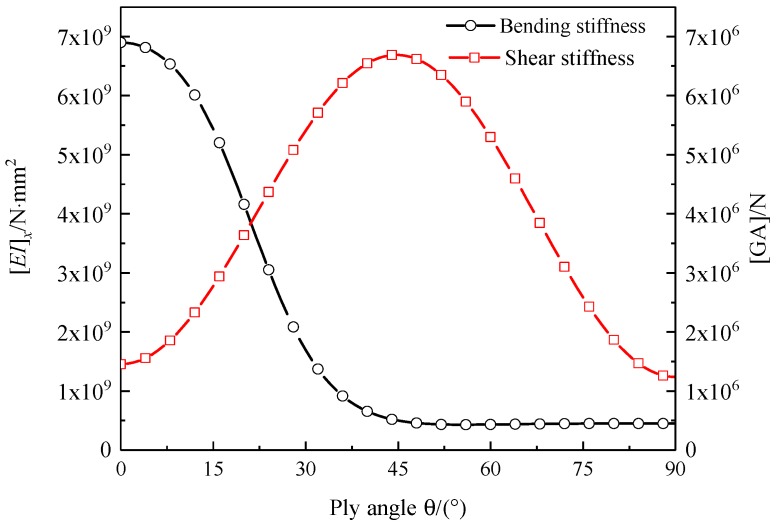
Bending stiffness [*EI*]*_x_* and shear stiffness [*GA*] of composite aluminum honeycomb sandwich laminated box beam changing with the ply angle.

**Figure 21 materials-12-02959-f021:**
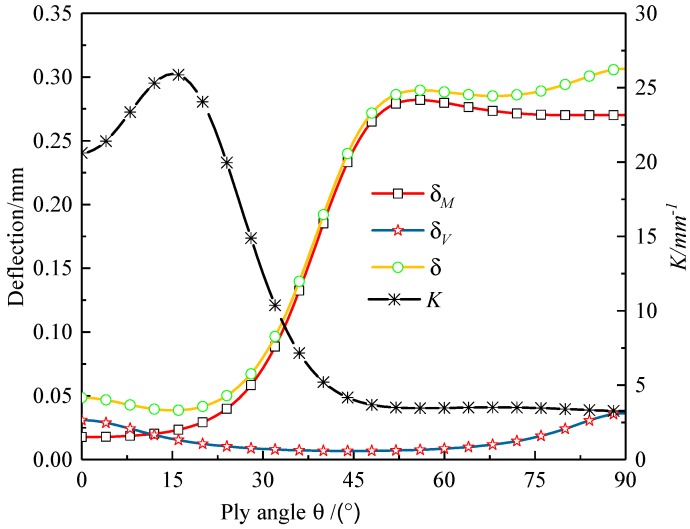
Deflection *δ* and stiffness coefficient *K* of composite aluminum honeycomb sandwich laminated box beam changing with the ply angle.

**Figure 22 materials-12-02959-f022:**
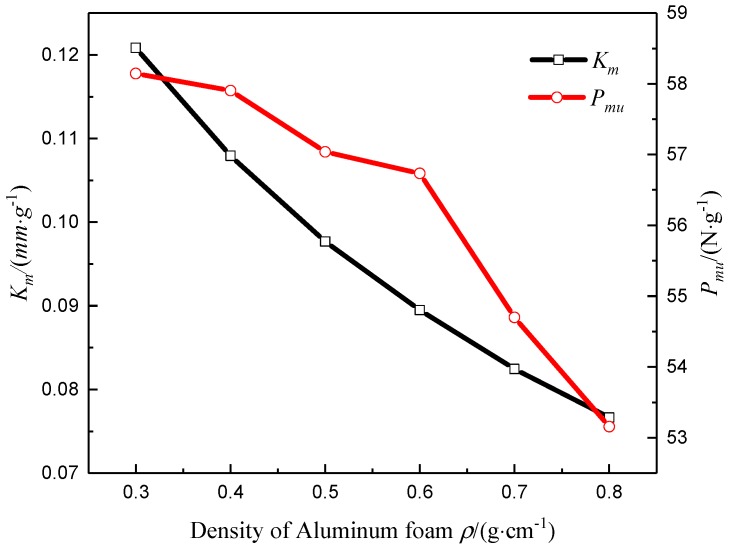
Specific stiffness parameter *K_m_* and specific load *P_mu_* changing with the density of aluminum foam.

**Figure 23 materials-12-02959-f023:**
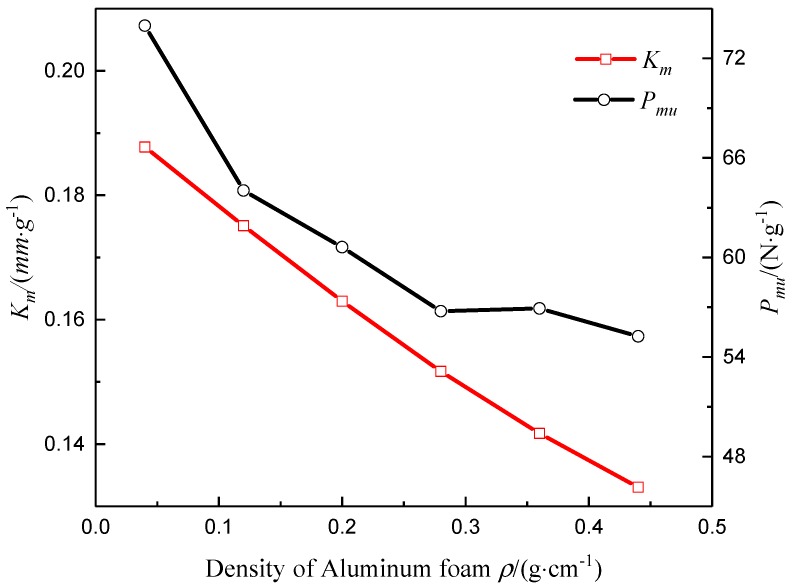
Specific stiffness *K_m_* and specific load *P_mu_* changing with the wall thickness of aluminum honeycomb.

**Table 1 materials-12-02959-t001:** The number and mass of the three different composite sandwich laminated box beams.

No.	Type of Core	Total Mass *m* (g)
B1	Aluminum honeycomb	109.4
B2	Aluminum foam	197.6
B3	Polyurethane elastomer	349.8

**Table 2 materials-12-02959-t002:** Bending experiment data of the composite sandwich laminated box beams.

No.	*K_b_* (N∙m^2^)	*P_u_* (N)	*K_mb_* (N∙m^2^/g)	*P_mu_* (N/g)
B1	3686.56	7590.15	32.42	69.51
B2	3707.98	11,211.87	21.25	56.74
B3	3693.66	6670.07	10.46	19.07

Note: *K_mb_* = *K_b_*/*m* is the ratio of bending stiffness to total mass; *P_mu_* = *P_u_*/*m*, is the ratio of ultimate load *P_u_* to total mass.

**Table 3 materials-12-02959-t003:** Material mechanical properties for the specimens.

Parameter	Value	Parameter	Value
*E*_1_ (GPa)	135	*X_t_* (MPa)	1673
*E*_2_*= E*_3_ (GPa)	8.8	*X_c_* (MPa)	1160
*G*_12_ = *G*_13_ (GPa)	4.47	*Y_t_* (MPa)	68
*G*_23_ (GPa)	3.0	*Y_c_* (MPa)	210
*v*_12_ = *v*_13_	0.33	*S_L_* = *S_T_* (MPa)	112
*v* _23_	0.33	*ρ* (g/cm^3^)	1.58

**Table 4 materials-12-02959-t004:** Composite material fracture energy parameters [8].

$G_{ft}^{C}\;(N/\mathrm{mm})$	$G_{fc}^{C}(N/\mathrm{mm})$	$G_{mt}^{C}\;(N/\mathrm{mm})$	$G_{mc}^{C}\;(N/\mathrm{mm})$
50.5	30.5	0.22	1.1

**Table 5 materials-12-02959-t005:** Material parameters of composite adhesive interface layer [13].

Parameter	Value	Parameter	Value
$t_{n}^{0}$ (MPa)	30	*K_nn_* (GPa)	1000
$t_{s}^{0}$ (MPa)	60	*K_ss_* (GPa)	1000
$t_{t}^{0}$ (MPa)	60	*K_tt_* (GPa)	1000
$G_{n}^{C}$ (N/mm)	0.2	*η*	1.5
$G_{s}^{C}$ (N/mm)	1.0	*ρ* (g/m^3^)	1.2
$G_{t}^{C}$ (N/mm)	1.002		

**Table 6 materials-12-02959-t006:** The theoretical and experimental bending stiffnesses of composite sandwich laminated box beams.

No.	Theoretical Calculation (N∙m^2^)	Experiment (N∙m^2^)	Error (%)
B1	3686.56	3495.26	−5.19
B2	3707.98	3529.02	−4.83
B3	3693.66	3544.75	−4.03

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
