# Peer review of "Bending Experiment and Mechanical Properties Analysis of Composite Sandwich Laminated Box Beams"

_materials, 2019, doi:10.3390/ma12182959_

Round 1

Reviewer 1 Report

Generally, the paper is of good quality and interesting.

Practically, I have one remark.

The Authors describe progressive damage analysis model.

I understand that the model is implemented into Abaqus program.

In my opinion, the Authors should clearly state that the constitutive model

is implemented there and explain the details of the implementation.

It  is the condition to publish the paper in the Journal.

Author Response

Dear reviewer:

Thank you for your valuable feedback! I have made serious and meticulous revisions based on your comments. The revised part has been marked red in the revised version. Please check it out. If you feel that there are other places that have not been modified, please let me know in time, I am very happy to make changes again!

I am graduating at the end of the year, but the number of papers I have published has not yet met the requirements of the school, so I especially hope that this manuscript can be accepted.

Thank you again for your review of this manuscript. I wish you a happy work. I look forward to your early reply.

The cover letter to explain point-by-point the details of the revisions in the manuscript and my responses to the reviewers' comments are as follows.

Generally, the paper is of good quality and interesting.

Practically, I have one remark.

The Authors describe progressive damage analysis model.

I understand that the model is implemented into Abaqus program.

In my opinion, the Authors should clearly state that the constitutive model is implemented there and explain the details of the implementation.

It is the condition to publish the paper in the Journal.

Reply:

 The constitutive model of the composite has been added and explained in Section 2.1.1,please check the part being marked red.

Reviewer 2 Report

The paper describes a numerical and experimental investigation of the linear and ultimate strength behavior of a composite beam subjected to three-point bending. Although the conclusions are somewhat obvious, the paper is suitable for the Journal in view of the analysis details described. Some comments have to be observed in order to improve the quality and clarity of the paper:
1. 'shear deformable beam theory': Apart from the fact, that the theory is not deformable, obviously Thmoshenko beam theory is meant.
2. 'stiffness coefficient' has to be defined. Later in Section 2.3.1 'bending equivalent stiffness' is used (and given in Table 6) without definition. Eq. (29) gives only 'bending stiffness'! Please explain better, e.g. what is meant with equivalent!
3. Section 1.1, line 7: Explain [0_4/90]_5!
4. Fig. 2 and Table 1: Use dimensions in figure (as these fixed).
5. 7 lines below Table 1: 'basically linear'. The reviewer cannot see linearity, but double-curved line.
6. It remains unclear what is meant by wing. Do you mean 'upper flange' (or 'upper plate')?
7. Fig. 4: Give values of displacements for 2nd and third load stage to see where we are in the load-displacement diagram.
8. Table 2, top line: P_u/N => P_u(N) to be in line with the other variables.
9. Text behind eq. (5): What means 'longitudinal' and 'transverse' shear? Not a coordinate, but a plane (two coordinates) determines shear!
10. Table 4: Why are the values given twice?
11. Element COH3D8 seems to refer to ABAQUS. Give reference and more explanation of properties for those who don't use ABAQUS.
12. Table 6: Units are missing.
13. Eq. (33): Maybe better mass per length?
14. Figs. 14 and 15: What is shown by the colors? Explain in detail! Legend not readable.
15. Section 3.1; The outcome is not really new and well-known.
16. 'stratified layer ratio' is unclear. Please explain in detail what is meant.
17. 'accurately' in the conclusions is not correct in view of the deviations!
18. Generally the reviewer wonders why the fibres have not been laid with 45 deg in view of shear forces acting.
Also the English wording and grammar requires modifications, as the following examples show. Furthermore, some sentences seem to be incomplete regarding subject, object and verb.
- composite sandwiched box beam: Do you mean sandwich box beam? The reviewer does to know a verb like 'to sandwich'.
- Heading of Section 1.1: What means 't'?
- Section 1.1, line 18: Delete one 'composite'.
- Section 1.2, line 3: 'measuring range of 100 kN': Do you mean load range?
- 12 lines below Table 1: start => starts.
- Note of Table 2: ration => ratio.
- Just before eq. (9): Colon given twice.
- Before eq. (13): cm3 => cm^3.
- Table 5: Check alignment of letters for last parameter.
- Behind eq. 26: double moment => bi-moment.
- Section 2.3.2: What is a cloud diagram? Shown are f.e. motels with deformed meshes and obviously stresses.
- Section 2.3.2, line 14: load displacement increases => load and displacement increase. 4 lines later: What means fo?
- Section 3.1, line 5: Insert 'and'.
- Figs. 16 and 19: Deflexion => Delection.
- Avoid bandworms like 'aluminum foam sandwich laminated box beam finite element model' (used in Conclusions, other bandworms throughout the text).
- References: Delete [J] etc. Avoid names in capital letters!
Check also the block format of paragraphs as well as spaces before opening brackets!

Author Response

Dear reviewer:

Thank you for your valuable feedback! I have made serious and meticulous revisions based on your comments. The revised part has been marked red in the revised version. Please check it out. If you feel that there are other places that have not been modified, please let me know in time, I am very happy to make changes again!

I am graduating at the end of the year, but the number of papers I have published has not yet met the requirements of the school, so I especially hope that this manuscript can be accepted.

Thank you again for your review of this manuscript. I wish you a happy work. I look forward to your early reply.

The cover letter to explain point-by-point the details of the revisions in the manuscript and my responses to the reviewers' comments are as follows.Please see the attachment.

The paper describes a numerical and experimental investigation of the linear and ultimate strength behavior of a composite beam subjected to three-point bending. Although the conclusions are somewhat obvious, the paper is suitable for the Journal in view of the analysis details described. Some comments have to be observed in order to improve the quality and clarity of the paper:

'shear deformable beam theory': Apart from the fact, that the theory is not deformable, obviously Thmoshenko beam theory is meant.

Reply:

All the 'shear deformable beam theory' has been replaced by 'Timoshenko beam theory', please check it.

'stiffness coefficient' has to be defined. Later in Section 2.3.1 'bending equivalent stiffness' is used (and given in Table 6) without definition. Eq. (29) gives only 'bending stiffness'! Please explain better, e.g. what is meant with equivalent!

Reply:

The 'stiffness coefficient' in Abstract has been replaced by ' the ratio of the overall stiffness to mass ',

The 'bending equivalent stiffness' in Section 2.3 has been replaced by 'bending stiffness'

The 'bending equivalent stiffness' is my clerical error.

Section 1.1, line 7: Explain [0_4/90]_5!

Reply:

The'[0_4/90]_5' is also a clerical error, which should be '[04/90]s '. '[04/90]s ' represents four layers of 0 degree and one layer of 90 degree, and the five layers are symmetric about the plane in the laminated wall. '[04/90]s '  is consistent with the representation of the laminate lay-up parameters in composite mechanics.

Fig. 2 and Table 1: Use dimensions in figure (as these fixed).

Reply:

  I have checked out Table 1 and Fig.2, the dimensions are consistent.

7 lines below Table 1: 'basically linear'. The reviewer cannot see linearity, but double-curved line.

Reply:

The 'basically linear' the load increases with the displacement

It remains unclear what is meant by wing. Do you mean 'upper flange' (or 'upper plate')?

Reply:

  Yes, the 'wing' should be 'upper flange'. All the 'wing' in the manuscript have been replaced.

Fig. 4: Give values of displacements for 2nd and third load stage to see where we are in the load-displacement diagram.

Reply:

The values of displacements of second and third load stage have been given on Fig.4

Table 2, top line: P_u/N => P_u(N) to be in line with the other variables.

Reply:

This minor mistake has been modified.

Text behind eq. (5): What means 'longitudinal' and 'transverse' shear? Not a coordinate, but a plane (two coordinates) determines shear!

Reply:

The 'longitudinal' and 'transverse' shear have been replaced by the in-plane shear.

Table 4: Why are the values given twice?

Reply:

  This is also a clerical error, which has been modified. The last two columns should be the values of Gmc and Gmt

Element COH3D8 seems to refer to ABAQUS. Give reference and more explanation of properties for those who don't use ABAQUS.

  Reply:

  Some explanations about COH3D8 have been added, as shown in Fig.11 and the above text. The more detailed introduction about COH3D8 can be found in Ref.[11].

Table 6: Units are missing.

  Reply:

  The units has been added.

Eq. (33): Maybe better mass per length?

  Reply:

  The lengths of the composite sandwich laminated box beam studied in Section 3.2 and Section 3.3 are fixed, so even if the mass in Eqs. (33) and (34) is replaced by the mass per length, the trend of curves in Figs.18/20/21/22 will not change, only the ordinate values and units will change.

 (Eqs. (33) and (34) correspond to Eqs. (37), (38) in the revised manuscript, respectively)

Figs. 14 and 15: What is shown by the colors? Explain in detail! Legend not readable.

  Reply:

  In the Fig.14 (Fig.16 in the revised manuscript), the darker the color, the more severe the damage state, and the red area indicates that elements are completely ineffective.

  In the Fig.15(Fig.17 in the revised manuscript), the darker the color, the more severe the stress concentration.

  All these explanation has been added

Section 3.1; The outcome is not really new and well-known.

  Reply:

  The authors do not reckon that the outcome in Section 3.1 are well known. Maybe the reviewer means that this outcome is well known for isotropic material beams, but this manuscript studies composite sandwich laminated box beams. The differences are as follows.

  For isotropic material beams, the elastic modulus, E, and shear modulus, G, satisfy the following relationship:

      (1)

If v = 0.3, the elastic modulus of elasticity is 2.6 times to the shear modulus. The two is still the same order of magnitude. When ratio of span to height is greater than 10, the effect of shear deformation is negligible because it is a small amount compared to bending deformation. Please refer to Materials Mechanics (the fifth edition, edited by Liu Hongwen) on page 78 for the detailed derivation of this conclusion.

  For a typical anisotropic material such as composite, the elastic modulus and shear modulus do not satisfy the Eq. (1). Take the data in Table 3 of the manuscript as an example:

Table 3 Material mechanical properties parameters for the example

Parameter

Value

Parameter

Value

E1/GPa

135

Xt/MPa

1673

E2= E3/GPa

8.8

Xc/MPa

1160

G12= G13/GPa

4.47

Yt /MPa

68

G23/GPa

3.0

Yc/MPa

210

v12= v13

0.33

SL= ST /MPa

112

v23

0.33

ρ/g/cm3

1.58

It can be found that E1 is 45 times to G23, and the two are not an order of magnitude. That is to say, the ability of the composite to resist shear deformation may be poor, and the shear deformation caused by the lateral concentrated load may be large. Therefore, the conclusion that "shear deformation is negligible" for isotropic materials does not necessarily apply to composite materials. This is a question that needs to be studied in depth.

In order to carry out the study, it is first assumed that the shear deformation of composite structures is also negligible. The verification is carried out by experiments, analytical models, and validating analytical models.

0 degree layers can improve the bending stiffness, and 45 degree layers can improve the shear stiffness. Assume that the shear deformation of composite structures is also negligible, the composite sandwich laminated box beam mainly undergoes bending deformation under concentrated load. Therefore, [04/90]s is selected as the lay-ups of specimens, which can give the specimen a greater bending stiffness. Moreover, the shear stiffness of the specimen of the [04/90]s lay-up is small, so the Eq. (31) can more clearly reflect the influence of the additional deflection caused by shear force. This also answers the 18th question that why the fibres have not been laid with 45 degrese.

The results of Eqs. (28) and (29) are used, and the quantitative analysis is performed based on the Eqs. (30) and (31). Finally, it is found that, the ratio of the shear deflection to the total deflection, decreases with the increase of the ratio of span to height, which is consistent with the isotropic material box beam. The assumption is true in a certain degree. However, composite sandwiched box beam is more affected by shear deformation. When the ratio of span to height is less than 10, the ratio of the shear deflection to the total deflection is above 25%, and for isotropic materials laminated box beam, that is only 3%..

Note:(Eq.(28), (29),(30),(31) correspond to Eqs.(32),(33),(34) and (35) in the revised manuscript, respectively)

'stratified layer ratio' is unclear. Please explain in detail what is meant.

Reply:

  The 'stratified layer ratio' has been replaced by ' the ratio of circumferential layers to longitudinal layers'. It is also a clerical error.

'accurately' in the conclusions is not correct in view of the deviations!

  Reply:

  The 'accurately' has been replaced by the ' approximately '.

Generally the reviewer wonders why the fibres have not been laid with 45 deg in view of shear forces acting.

Reply:

The reason is explained in the reply to question 16. The problem that the 45-degree layers can improve the local buckling load is an object in my another paper.

Also the English wording and grammar requires modifications, as the following examples show. Furthermore, some sentences seem to be incomplete regarding subject, object and verb.- composite sandwiched box beam: Do you mean sandwich box beam? The reviewer does to know a verb like 'to sandwich'.

- Heading of Section 1.1: What means 't'?

- Section 1.1, line 18: Delete one 'composite'.

- Section 1.2, line 3: 'measuring range of 100 kN': Do you mean load range?

- 12 lines below Table 1: start => starts.

- Note of Table 2: ration => ratio.

- Just before eq. (9): Colon given twice.

- Before eq. (13): cm3 => cm^3.

- Table 5: Check alignment of letters for last parameter.

- Behind eq. 26: double moment => bi-moment.

- Section 2.3.2: What is a cloud diagram? Shown are f.e. motels with deformed meshes and obviously stresses.

- Section 2.3.2, line 14: load displacement increases => load and displacement increase. 4 lines later: What means fo?

- Section 3.1, line 5: Insert 'and'.

- Figs. 16 and 19: Deflexion => Delection.

- Avoid bandworms like 'aluminum foam sandwich laminated box beam finite element model' (used in Conclusions, other bandworms throughout the text).

- References: Delete [J] etc. Avoid names in capital letters! Check also the block format of paragraphs as well as spaces before opening brackets!

Reply:

  All the mistakes mentioned above has been modified, and the authors have the revised manuscript checked by native English speaker.

Round 2

Reviewer 1 Report

Dear Authors,

The Authors introduced the amendments.

The quality of the paper is increased.

In my opinion, the paper can appear in the Journal.

Author Response

Dear reviewerï¼›

  Thank you very much.

  Best regards and wishes to you

Reviewer 2 Report

The authors have considered all reviewer's comments. The following additional comments are given, the numbers referring to the original comments:

4. The reviewer's proposal was to include the actual dimensions in the figure in order to make it easier for the reader to understand the structure. Table 1 can then be reduced to the first two and the last column.
14. The 'damage' shown in Fig. 16 should be defined somewhere (can it be both, matrix and fibre failure?). However, the darkest area is blue, which affects regions being not damaged. Please clarify and revise explanation. Fig. 17 seems to show von Mises stresses (in the matrix?), which has to be mentioned in the figure caption.

Author Response

Dear reviewer:

Thank you for your valuable comments. I have revised and improved the manuscript based on your comments, as shown below.

All revisions has been be clearly highlighted with red bold font.

If you feel there is anything that has not been modified, please let me know and I will be happy to change it again.

The reviewer's proposal was to include the actual dimensions in the figure in order to make it easier for the reader to understand the structure. Table 1 can then be reduced to the first two and the last column.

Reply:

Fig. 2 and Table 1 have been revised based on the reviewer's comments, please check.

The 'damage' shown in Fig. 16 should be defined somewhere (can it be both, matrix and fibre failure?). However, the darkest area is blue, which affects regions being not damaged. Please clarify and revise explanation. Fig. 17 seems to show von Mises stresses (in the matrix?), which has to be mentioned in the figure caption.

  Reply:

  The approximate area of the damage in Fig.16 has been marked by a red circle. Fig.16 shows four modes of failure of composite sandwiched box girder under ultimate load. These four modes occur simultaneously under load and simultaneously, but the location and extent of their occurrence are different.

The finite element model shows the location and extent of these four modes separately. There are more informations on the modes relevant criteria of the Hashin in Ref.[11] and [14].

The legend of Fig. 16 can still be seen clearly, from blue to red, the degree of damage is becoming more and more serious. The author's statement in the first revised manuscript is not clear enough.

Fig. 17 is the stress clouds of the entire composite sandwich laminated box beam. The captipn of figure has been modified, please check it.

I am graduating at the end of the year, but the number of papers I have published has not yet met the requirements of the school, so I very much hope that this paper can be accepted.

Once again, I would like to extend my sincere gratitude to you.

Best regards
